# Training language models to follow instructions with human feedback

**Long Ouyang**[*]     **Jeff Wu**[*]     **Xu Jiang**[*]     **Diogo Almeida**[*]     **Carroll L. Wainwright**[*]

**Pamela Mishkin**[*]     **Chong Zhang**     **Sandhini Agarwal**     **Katarina Slama**     **Alex Ray**

**John Schulman**     **Jacob Hilton**     **Fraser Kelton**     **Luke Miller**     **Maddie Simens**

**Amanda Askell**[†]     **Peter Welinder**     **Paul Christiano**[*†]

**Jan Leike**[*]     **Ryan Lowe**[*]

OpenAI

## Abstract

Making language models bigger does not inherently make them better at following a user's intent. For example, large language models can generate outputs that are untruthful, toxic, or simply not helpful to the user. In other words, these models are not *aligned* with their users. In this paper, we show an avenue for aligning language models with user intent on a wide range of tasks by fine-tuning with human feedback. Starting with a set of labeler-written prompts and prompts submitted through a language model API, we collect a dataset of labeler demonstrations of the desired model behavior, which we use to fine-tune GPT-3 using supervised learning. We then collect a dataset of rankings of model outputs, which we use to further fine-tune this supervised model using reinforcement learning from human feedback. We call the resulting models *InstructGPT*. In human evaluations on our prompt distribution, outputs from the 1.3B parameter InstructGPT model are preferred to outputs from the 175B GPT-3, despite having 100x fewer parameters. Moreover, InstructGPT models show improvements in truthfulness and reductions in toxic output generation while having minimal performance regressions on public NLP datasets. Even though InstructGPT still makes simple mistakes, our results show that fine-tuning with human feedback is a promising direction for aligning language models with human intent.

## 1   Introduction

Large language models (LMs) can be prompted to perform a range of natural language processing (NLP) tasks, given some examples of the task as input. However, these models often express unintended behaviors such as making up facts, generating biased or toxic text, or simply not following user instructions (Bender et al., 2021; Bommasani et al., 2021; Kenton et al., 2021; Weidinger et al., 2021; Tamkin et al., 2021; Gehman et al., 2020). This is because the language modeling objective

---

[*]Primary authors. This was a joint project of the OpenAI Alignment team. RL and JL are the team leads. Corresponding author: `lowe@openai.com`.

[†]Work done while at OpenAI. Current affiliations: AA: Anthropic; PC: Alignment Research Center.

36th Conference on Neural Information Processing Systems (NeurIPS 2022).

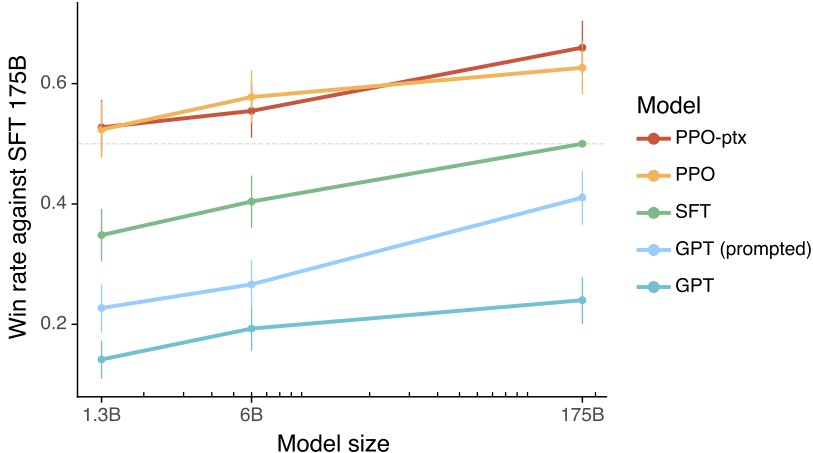

Figure 1: Human evaluations of various models on the API prompt distribution, evaluated by how often outputs from each model were preferred to those from the 175B SFT model. Our InstructGPT models (PPO-ptx) as well as its variant trained without pretraining mix (PPO) significantly outperform the GPT-3 baselines (GPT, GPT prompted); outputs from our 1.3B PPO-ptx model are preferred to those from the 175B GPT-3. Error bars throughout the paper are 95% confidence intervals.

used for many recent large LMs—predicting the next token on a webpage from the internet—is different from the objective "follow the user's instructions helpfully and safely" (Radford et al., 2019; Brown et al., 2020; Fedus et al., 2021; Rae et al., 2021; Thoppilan et al., 2022). Thus, we say that the language modeling objective is *misaligned*. Averting these unintended behaviors is especially important for language models that are deployed and used in hundreds of applications.

We make progress on aligning language models by training them to act in accordance with the user's intention (Leike et al., 2018). This encompasses both explicit intentions such as following instructions and implicit intentions such as staying truthful, and not being biased, toxic, or otherwise harmful. Using the language of Askell et al. (2021), we want language models to be *helpful* (they should help the user solve their task), *honest* (they shouldn't fabricate information or mislead the user), and *harmless* (they should not cause physical, psychological, or social harm to people or the environment). We elaborate on the evaluation of these criteria in Section 3.5.

We focus on *fine-tuning* approaches to aligning language models. Specifically, we use reinforcement learning from human feedback (RLHF; Christiano et al., 2017; Stiennon et al., 2020) to fine-tune GPT-3 to follow a broad class of written instructions (see Figure 2). This technique uses human preferences as a reward signal to fine-tune our models. We first hire a team of 40 contractors to label our data, based on their performance on a screening test (see Section 3.3 and Appendix B.1 for more details). We then collect a dataset of human-written demonstrations of the desired output behavior on (mostly English) prompts submitted to a language model API and some labeler-written prompts, and use this to train our supervised learning baselines. Next, we collect a dataset of human-labeled comparisons between outputs from our models on a larger set of API prompts. We then train a reward model (RM) on this dataset to predict which model output our labelers would prefer. Finally, we use this RM as a reward function and fine-tune our supervised learning baseline to maximize this reward using the PPO algorithm (Schulman et al., 2017). We illustrate this process in Figure 2. This procedure aligns the behavior of GPT-3 to the stated preferences of a specific group of people (mostly our labelers and researchers), rather than any broader notion of "human values"; we discuss this further in Appendix G.2. We call the resulting models *InstructGPT*.

We mainly evaluate our models by having our labelers rate the quality of model outputs on our test set, consisting of prompts from held-out users (who are not represented in the training data). We also conduct automatic evaluations on a range of public NLP datasets. We train three model sizes (1.3B, 6B, and 175B parameters), and all of our models use the GPT-3 architecture. Our main findings are:

**Labelers significantly prefer InstructGPT outputs over outputs from GPT-3.** Outputs from the 1.3B parameter InstructGPT model are preferred to outputs from the 175B GPT-3, despite having

over 100x fewer parameters. These models have the same architecture, and differ only by the fact that InstructGPT is fine-tuned on our human data. This result holds true even when we add a few-shot prompt to GPT-3 to make it better at following instructions. Outputs from our 175B InstructGPT are preferred to 175B GPT-3 outputs $85 \pm 3\%$ of the time, and preferred $71 \pm 4\%$ of the time to few-shot 175B GPT-3. InstructGPT also generates more appropriate outputs according to our labelers.

**InstructGPT models show improvements in truthfulness over GPT-3.** On the TruthfulQA benchmark, InstructGPT generates truthful and informative answers more often than GPT-3. On "closed-domain" tasks from our API prompt distribution, where the output should not contain information that is not present in the input, InstructGPT models make up information not present in the input about half as often as GPT-3 (a 21% vs. 41% hallucination rate, respectively).

**InstructGPT shows small improvements in toxicity over GPT-3, but not bias.** To measure toxicity, we use the RealToxicityPrompts dataset (Gehman et al., 2020) and conduct both automatic and human evaluations. InstructGPT models generate about 25% fewer toxic outputs than GPT-3 when prompted to be respectful. InstructGPT does not significantly improve over GPT-3 on the Winogender (Rudinger et al., 2018) and CrowSPairs (Nangia et al., 2020) datasets.

**We can minimize performance regressions on public NLP datasets by modifying our RLHF fine-tuning procedure.** During RLHF fine-tuning, we observe performance regressions compared to GPT-3 on certain public NLP datasets. We can greatly reduce the performance regressions on these datasets by mixing PPO updates with updates that increase the log likelihood of the pretraining distribution (PPO-ptx), without compromising labeler preference scores.

**Our models generalize to the preferences of "held-out" labelers that did not produce any training data.** To test the generalization of our models, we conduct a preliminary experiment with held-out labelers, and find that they prefer InstructGPT outputs to outputs from GPT-3 at about the same rate as our training labelers. However, more work is needed to study how these models perform on broader groups of users, and how they perform on inputs where humans disagree about the desired behavior.

**Public NLP datasets are not reflective of how our language models are used.** We compare GPT-3 fine-tuned on our human preference data (i.e. InstructGPT) to GPT-3 fine-tuned on two different compilations of public NLP tasks: the FLAN (Wei et al., 2021) and T0 (Sanh et al., 2021) (in particular, the T0++ variant). These datasets consist of a variety of NLP tasks, combined with natural language instructions for each task. On our API prompt distribution, our FLAN and T0 models perform slightly worse than our SFT baseline, and labelers significantly prefer InstructGPT to these models.

**InstructGPT models show promising generalization to instructions outside of the RLHF fine-tuning distribution.** We qualitatively probe InstructGPT's capabilities, and find that it is able to follow instructions for summarizing code, answer questions about code, and sometimes follows instructions in different languages, despite these instructions being very rare in the fine-tuning distribution. This result is exciting because it suggests that our models are able to generalize the notion of "following instructions." They retain some alignment even on tasks for which they get very little direct supervision.

**InstructGPT still makes simple mistakes.** For example, InstructGPT can still fail to follow instructions, make up facts, give long hedging answers to simple questions, or fail to detect instructions with false premises.

Overall, our results indicate that fine-tuning large language models using human preferences significantly improves their behavior on a wide range of tasks, though much work remains to be done to improve their safety and reliability.

## 2 Related work

**Research on alignment and learning from human feedback.** We build on previous techniques to align models with human intentions, particularly reinforcement learning from human feed-

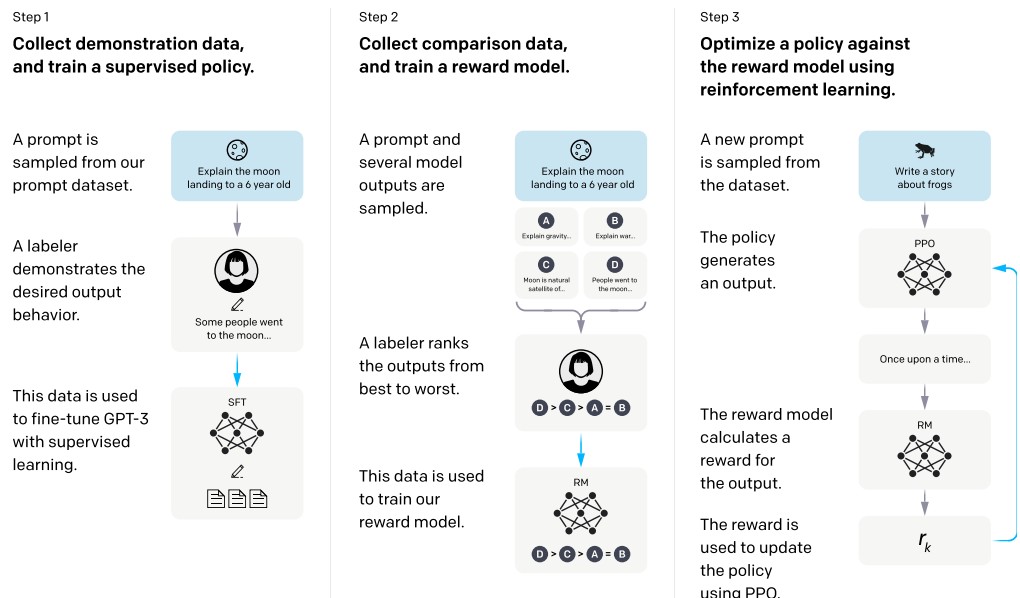

Figure 2: A diagram illustrating the three steps of our method: (1) supervised fine-tuning (SFT), (2) reward model (RM) training, and (3) reinforcement learning via proximal policy optimization (PPO) on this reward model. Blue arrows indicate that this data is used to train one of our models. In Step 2, boxes A-D are samples from our models that get ranked by labelers.

back (RLHF). Originally developed for training simple robots in simulated environments and Atari games (Christiano et al., 2017; Ibarz et al., 2018), it has recently been applied to fine-tuning language models to summarize text (Ziegler et al., 2019; Stiennon et al., 2020; Böhm et al., 2019; Wu et al., 2021). This work is in turn influenced by similar work using human feedback as a reward in domains such as dialogue (Jaques et al., 2019; Yi et al., 2019; Hancock et al., 2019), translation (Kreutzer et al., 2018; Bahdanau et al., 2016), semantic parsing (Lawrence and Riezler, 2018), story generation (Zhou and Xu, 2020), review generation (Cho et al., 2018), and evidence extraction (Perez et al., 2019). In concurrent work, Askell et al. (2021); Bai et al. (2022) propose language assistants as a testbed for alignment research, and train models using RLHF. Our work can be seen as a direct application of RLHF to aligning language models on a broad distribution of language tasks.

**Training language models to follow instructions.** Our work is also related to research on cross-task generalization in language models, where LMs are fine-tuned on a broad range of public NLP datasets (usually prefixed with an appropriate instruction) and evaluated on a different set of NLP tasks. There has been a range of work in this domain (Yi et al., 2019; Mishra et al., 2021; Wei et al., 2021; Khashabi et al., 2020; Sanh et al., 2021; Aribandi et al., 2021), which differ in training and evaluation data, formatting of instructions, size of pretrained models, and other experimental details.

**Mitigating the harms of language models.** A goal of modifying the behavior of language models is to mitigate the harms of these models when they're deployed in the real world. These risks have been extensively documented (Bender et al., 2021; Bommasani et al., 2021; Kenton et al., 2021; Weidinger et al., 2021; Tamkin et al., 2021). Language models can produce biased outputs (Dhamala et al., 2021; Liang et al., 2021; Manela et al., 2021; Caliskan et al., 2017; Kirk et al., 2021), leak private data (Carlini et al., 2021), generate misinformation (Solaiman et al., 2019; Buchanan et al., 2021), and be used maliciously; for a thorough review we direct the reader to Weidinger et al. (2021). There are many ways to mitigate these harms, including by fine-tuning on a small, value-targeted dataset (Solaiman and Dennison, 2021), filtering the pretraining dataset (Ngo et al., 2021), or human-in-the-loop data collection (Dinan et al., 2019; Xu et al., 2020).

# 3 Methods and experimental details

## 3.1 High-level methodology

Our methodology follows that of Ziegler et al. (2019) and Stiennon et al. (2020), who applied it in the stylistic continuation and summarization domains. We start with a pretrained language model (Radford et al., 2019; Brown et al., 2020; Fedus et al., 2021; Rae et al., 2021; Thoppilan et al., 2022), a distribution of prompts on which we want our model to produce aligned outputs, and a team of trained human labelers (see Section 3.3 for details). We then apply the following three steps (Figure 2).

**Step 1: Collect demonstration data, and train a supervised policy.** Our labelers provide demonstrations of the desired behavior on the input prompt distribution (see Section 3.2 for details on this distribution). We then fine-tune a pretrained GPT-3 model on this data using supervised learning.

**Step 2: Collect comparison data, and train a reward model.** We collect a dataset of comparisons between model outputs, where labelers indicate which output they prefer for a given input. We then train a reward model to predict the human-preferred output.

**Step 3: Optimize a policy against the reward model using PPO.** We use the output of the RM as a scalar reward. We fine-tune the supervised policy to optimize this reward using the PPO algorithm (Schulman et al., 2017).

Steps 2 and 3 can be iterated continuously; more comparison data is collected on the current best policy, which is used to train a new RM and then a new policy. In practice, most of our comparison data comes from our supervised policies, with some coming from our PPO policies.

## 3.2 Dataset

Our prompt dataset consists primarily of text prompts submitted to a commercial language model API, as well as a small number of labeler-written prompts. These prompts are very diverse and include generation, question answering, dialog, summarization, extractions, and other natural language tasks (see Appendix A). Our dataset is over 96% English. We heuristically deduplicate prompts, and ensure that the validation and test sets contain no data from users whose data is in the training set. We also filter prompts containing personally identifiable information (PII).

From these prompts, we produce three different datasets used in our fine-tuning procedure: (1) our SFT dataset, with labeler demonstrations used to train our SFT models, (2) our RM dataset, with labeler rankings of model outputs used to train our RMs, and (3) our PPO dataset, without any human labels, which are used as inputs for RLHF fine-tuning. The SFT dataset contains about 13k training prompts (from the API and labeler-written), the RM dataset has 33k training prompts (from the API and labeler-written), and the PPO dataset has 31k training prompts (only from the API). More details on dataset sizes are provided in Table 3.

## 3.3 Human data collection

To produce our demonstration and comparison data, and to conduct our main evaluations, we hired a team of about 40 contractors on Upwork and through ScaleAI. Compared to earlier work that collects human preference data on the task of summarization (Ziegler et al., 2019; Stiennon et al., 2020; Wu et al., 2021), our inputs span a much broader range of tasks, and can occasionally include controversial and sensitive topics. Our aim was to select a group of labelers who were sensitive to the preferences of different demographic groups, and who were good at identifying outputs that were potentially harmful. Thus, we conducted a screening test designed to measure labeler performance on these axes (see Appendix B.1). As an initial study to see how well our model generalizes to the preferences of other labelers, we hire a separate set of labelers who do not produce any of the training data. These labelers are sourced from the same vendors, but do not undergo a screening test.

Despite the complexity of the task, we find that inter-annotator agreement rates are quite high: training labelers agree with each-other $72.6 \pm 1.5\%$ of the time, while for held-out labelers this number is $77.3 \pm 1.3\%$. For comparison, in the summarization work of Stiennon et al. (2020) researcher-researcher agreement was $73 \pm 4\%$.

## 3.4 Models

Starting from GPT-3 (Brown et al., 2020), we train models with three different techniques:

**Supervised fine-tuning (SFT).**   We fine-tune GPT-3 on our labeler demonstrations using supervised learning. We trained for 16 epochs, using a cosine learning rate decay, and residual dropout of 0.2. We do our final SFT model selection based on the RM score on the validation set. Similarly to Wu et al. (2021), we find that our SFT models overfit on validation loss after 1 epoch; however, we find that training for more epochs helps both the RM score and human preference ratings.

**Reward modeling (RM).**   We fine-tune GPT-3 to take in a prompt and response, and output a scalar reward. In this paper we only use 6B RMs, as this saves a lot of compute, and we found that 175B RM training could be unstable and thus was less suitable to be used as the value function during RL (see Appendix D for more details).

In Stiennon et al. (2020), the RM is trained on a dataset of comparisons between two model outputs on the same input. They use a cross-entropy loss, with the comparisons as labels—the difference in rewards represents the log odds that one response will be preferred to the other by a human labeler. In order to speed up comparison collection, we have labelers rank between $K = 4$ and $K = 9$ responses, and train on all $\binom{K}{2}$ comparisons from each prompt as a single batch element, for computational efficiency (see Appendix D. The loss function for the RM becomes:

$$\text{loss}\,(\theta) = -\frac{1}{\binom{K}{2}} E_{(x,y_w,y_l)\sim D}\left[\log\left(\sigma\left(r_\theta\left(x, y_w\right) - r_\theta\left(x, y_l\right)\right)\right)\right] \tag{1}$$

where $r_\theta(x, y)$ is the scalar output of the reward model for prompt $x$ and completion $y$ with parameters $\theta$, $y_w$ is the preferred completion out of the pair of $y_w$ and $y_l$, and $D$ is the comparison dataset.

**Reinforcement learning (RL).**   Again following Stiennon et al. (2020), we fine-tuned the SFT model using PPO (Schulman et al., 2017). The environment is a bandit environment which presents a random user prompt and expects a response to the prompt. Given the prompt and response, it produces a reward determined by the reward model and ends the episode. In addition, we add a per-token KL penalty from the SFT model at each token to mitigate over-optimization of the reward model. The value function is initialized from the RM. We call these models "PPO."

We also experiment with mixing the pretraining gradients into the PPO gradients, in order to fix the performance regressions on public NLP datasets (see Appendix D.4). We call these models "PPO-ptx." Unless otherwise specified, in this paper InstructGPT refers to the PPO-ptx models.

**Baselines.**   We compare the performance of our PPO models to our SFT models and GPT-3. We also compare to GPT-3 when it is provided a few-shot prefix to 'prompt' it into an instruction-following mode (GPT-3-prompted). This prefix is prepended to the user-specified instruction.

We additionally compare InstructGPT to fine-tuning 175B GPT-3 on the FLAN (Wei et al., 2021) and T0 (Sanh et al., 2021) datasets, which both consist of a variety of NLP tasks, combined with natural language instructions for each task (they differ in the NLP datasets included, and the style of instructions used). We fine-tune them on approximately 1 million examples and choose the checkpoint which obtains the highest RM score on the validation set (see Appendix D for more details).

## 3.5 Evaluation

Following Askell et al. (2021), we say our models are aligned if they are helpful, truthful, and harmless (we elaborate in Appendix C.2). We divide our quantitative evaluations into two parts:

**Evaluations on API distribution.**   Our main metric is human preference ratings on a held out set of prompts from the same source as our training distribution. When using prompts from the API for evaluation, we only select prompts by users we haven't included in training. For each model we calculate how often its outputs are preferred to a baseline policy; we choose our 175B SFT model as the baseline since its performance is near the middle of the pack. Additionally, we ask labelers to judge the overall quality of each response on a 1-7 Likert scale and collect a range of metadata for each model output (see Table 11). In particular, we collect data that aims to capture different

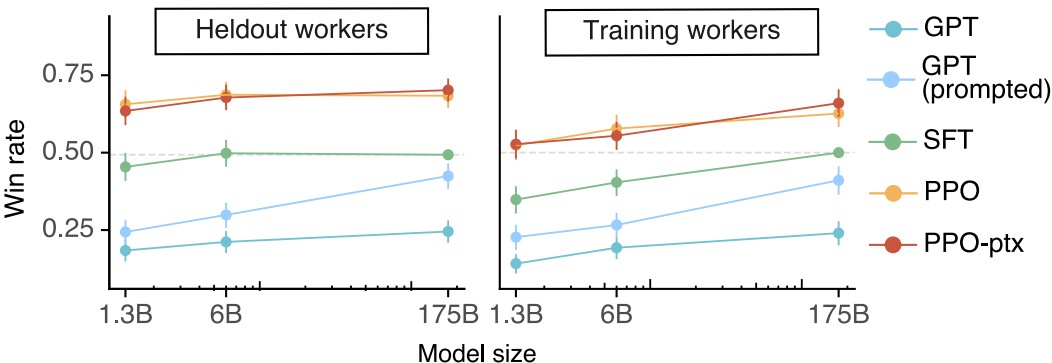

Figure 3: Preference results of our models, measured by winrate against the 175B SFT model.

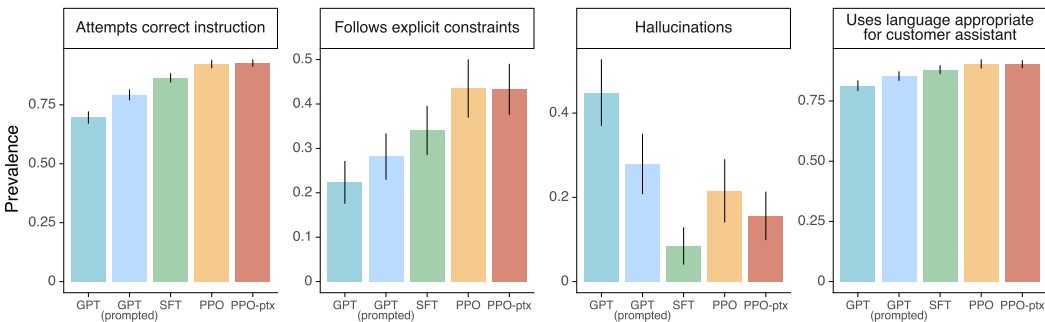

Figure 4: Metadata results on the API distribution, averaged over model sizes.

aspects of behavior in a deployed model that could end up being harmful: we have labelers evaluate whether an output is inappropriate in the context of a customer assistant, denigrates a protected class, or contains sexual or violent content.

**Evaluations on public NLP datasets.** We evaluate on two types of public datasets: those that capture an aspect of language model safety, particularly truthfulness, toxicity, and bias, and those that capture zero-shot performance on traditional NLP tasks like question answering, reading comprehension, and summarization. We also conduct human evaluations on the RealToxicityPrompts dataset (Gehman et al., 2020).

# 4 Results

## 4.1 Results on the API distribution

**Labelers significantly prefer InstructGPT outputs over outputs from GPT-3.** On our test set, our labelers significantly prefer InstructGPT outputs across model sizes (Figure 1). We find that GPT-3 outputs perform the worst, and one can obtain significant step-size improvements by using a well-crafted few-shot prompt (GPT-3 (prompted)), then by training on demonstrations using supervised learning (SFT), and finally by training on comparison data using PPO. Adding updates on the pretraining mix during PPO does not lead to large changes in labeler preference. To illustrate the magnitude of our gains: when compared directly, 175B InstructGPT outputs are preferred to GPT-3 outputs 85 ± 3% of the time, and preferred 71 ± 4% of the time to few-shot GPT-3.

In Figure 4 we show that labelers also rate InstructGPT outputs favorably along several more concrete axes. Specifically, compared to GPT-3, InstructGPT outputs are more appropriate in the context of a customer assistant, more often follow explicit constraints defined in the instruction (e.g. "Write your answer in 2 paragraphs or less."), are less likely to fail to follow the correct instruction entirely, and make up facts ('hallucinate') less often in closed-domain tasks.

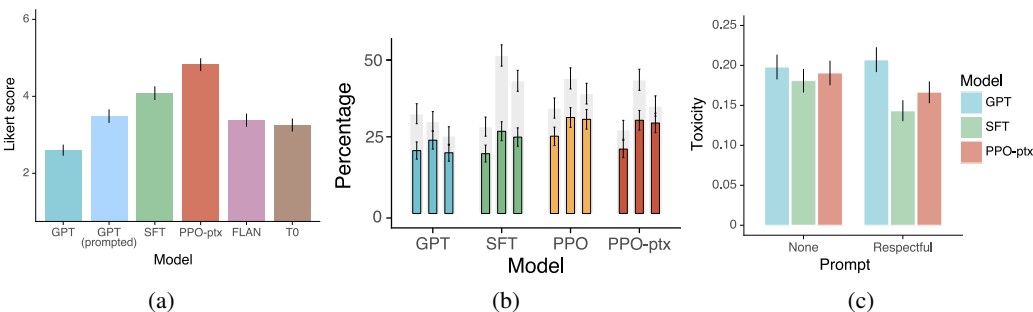

Figure 5: (a) Comparing our models with GPT-3 fine-tuned on the FLAN and T0 datasets, in terms of 1-7 Likert scores, on our prompt distribution. (b) Human evaluations on the TruthfulQA dataset. Gray bars indicate ratings of truthfulness; colored bars indicate ratings of truthfulness *and* informativeness. (c) Human evaluations on RealToxicityPrompts, with and without "respectful" instructions.

**Our models generalize to the preferences of "held-out" labelers that did not produce any training data.** Held-out labelers have similar ranking preferences as workers who we used to produce training data (see Figure 3). In particular, according to held-out workers, all of our InstructGPT models still greatly outperform the GPT-3 baselines. Thus, our InstructGPT models aren't simply overfitting to the preferences of our training labelers.

**Public NLP datasets are not reflective of how our language models are used.** In Figure 5a, we also compare InstructGPT to our 175B GPT-3 baselines fine-tuned on the FLAN (Wei et al., 2021) and T0 (Sanh et al., 2021) datasets (see Appendix D for details). We find that these models perform better than GPT-3, on par with GPT-3 with a well-chosen prompt, and worse than our SFT baseline. This indicates that these datasets are not sufficiently diverse to improve performance on our API prompt distribution. We believe this is partly because academic datasets focus on tasks where performance is easily measured, like classification and QA, while our API distribution consists of mostly (about 57%) open-ended generation tasks.

### 4.2 Results on public NLP datasets

**InstructGPT models show improvements in truthfulness over GPT-3.** As measured by human evaluations on the TruthfulQA dataset, our PPO models show small but significant improvements in generating truthful and informative outputs compared to GPT-3 (see Figure 5b). This behavior is the default: our models do not have to be specifically instructed to tell the truth to exhibit improved truthfulness. Interestingly, the exception is our 1.3B PPO-ptx model, which performs slightly worse than a GPT-3 model of the same size. Our improvements in truthfulness are also evidenced by the fact that our PPO models hallucinate less often on closed-domain tasks (Figure 4).

**InstructGPT shows small improvements in toxicity over GPT-3, but not bias.** We first evaluate our models on the RealToxicityPrompts dataset (Gehman et al., 2020) using human evaluations. Our results are in Figure 5c. We find that, when instructed to produce a safe and respectful output ("respectful prompt"), InstructGPT models generate less toxic outputs than those from GPT-3 according to the Perspective API. This advantage disappears when the respectful prompt is removed ("no prompt"). We see similar results when evaluating using the Perspective API (Appendix F.7).

**We can minimize performance regressions on public NLP datasets by modifying our RLHF fine-tuning procedure.** In Figure 25 we show that adding pretraining updates to our PPO fine-tuning (PPO-ptx) mitigates performance regressions on public NLP datasets, and even surpasses GPT-3 on HellaSwag. The performance of the PPO-ptx model still lags behind GPT-3 on DROP, SQuADv2, and translation; more work is needed to study and further eliminate these performance regressions. We also find that mixing in pretraining updates performs better than the simpler solution of increasing the KL coefficient (Figure 36).

### 4.3 Qualitative results

**InstructGPT models show promising generalization to instructions outside of the RLHF fine-tuning distribution.** In particular, we find that InstructGPT shows ability to follow instructions in non-English languages, and perform summarization and question-answering for code. This is interesting because non-English languages and code form a tiny minority of our fine-tuning data, and it suggests that, in some cases, alignment methods could generalize to producing the desired behavior on inputs that humans did not directly supervise. We show some qualitative examples in Figure 26.

**InstructGPT still makes simple mistakes.** In interacting with our 175B PPO-ptx model, we have noticed it can still make simple mistakes, despite its strong performance on many different language tasks. To give a few examples: (1) when given an instruction with a false premise, the model sometimes incorrectly assumes the premise is true, (2) the model can overly hedge; when given a simple question, it can sometimes say that there is no one answer to the question and give multiple possible answers, even when there is one fairly clear answer from the context, and (3) the model's performance degrades when instructions contain multiple explicit constraints (e.g. "list 10 movies made in the 1930's set in France") or when constraints can be challenging for language models (e.g. writing a summary in a specified number of sentences).

We show some examples of these behaviors in Figure 27. We suspect that behavior (2) emerges partly because we instruct labelers to reward epistemic humility; thus, they may tend to reward outputs that hedge, and this gets picked up by our reward model. We suspect that behavior (1) occurs because there are few prompts in the training set that assume false premises, and our models don't generalize well to these examples. We believe both these behaviors could be dramatically reduced with adversarial data collection (Dinan et al., 2019).

## 5 Discussion

### 5.1 Implications for alignment research

Our approach to alignment research in this work is iterative: we are improving the alignment of current AI systems instead of focusing abstractly on aligning AI systems that don't yet exist, which provides us with a clear empirical feedback loop of what works and what does not. We believe that this feedback loop is essential to refine our alignment techniques, and it forces us to keep pace with progress in machine learning.

From this work, we can draw lessons for alignment research more generally. First, the cost of increasing model alignment is modest relative to pretraining. Training our 175B SFT model requires 4.9 petaflops/s-days and training our 175B PPO-ptx model requires 60 petaflops/s-days, compared to 3,640 petaflops/s-days for GPT-3 (Brown et al., 2020). At the same time, our results show that RLHF is very effective at making language models more helpful to users, more so than a 100x model size increase. This suggests that right now increasing investments in alignment of existing language models is more cost-effective than training larger models. Second, we've seen some evidence that InstructGPT generalizes 'following instructions' to settings that we don't supervise it in. This is an important property because it's prohibitively expensive to have humans supervise models on every task they perform. Finally, we were able to mitigate most of the performance degradations introduced by our fine-tuning. If this was not the case, these performance degradations would constitute an alignment tax—an additional cost for aligning the model. Any alignment technique with a high tax might not see adoption, and thus such a tax is important to avoid.

### 5.2 Limitations

**Methodology.** The behavior of our InstructGPT models is determined in part by the human feedback obtained from our contractors. Some of the labeling tasks rely on value judgments that may be impacted by the identity of our contractors, their beliefs, cultural backgrounds, and personal history. We kept our team of contractors small because this facilitates high-bandwidth communication with a smaller set of contractors who are doing the task full-time. However, this group is clearly not representative of the full spectrum of people affected by these models. As a simple example, our labelers are primarily English-speaking and our data consists almost entirely of English instructions.

**Models.** Our models are neither fully aligned nor fully safe; they still generate toxic or biased outputs, make up facts, and generate sexual and violent content without explicit prompting. They can also fail to generate reasonable outputs on some inputs; we show some examples of this in Figure 27. Perhaps the greatest limitation of our models is that, in most cases, they follow the user's instruction, even if that could lead to harm in the real world. For example, when prompting the models to be maximally biased, InstructGPT generates more toxic outputs than equivalently-sized GPT-3 models.

## 5.3 Broader impacts

This work is motivated by our aim to increase the positive impact of large language models by training them to do what a given set of humans want them to do. By default, language models optimize the next word prediction objective, which is only a proxy for what we want these models to do. Our results indicate that our techniques hold promise for making language models more helpful, truthful, and harmless. In the longer term, alignment failures could lead to more severe consequences, particularly if these models are deployed in safety-critical situations.

However, making language models better at following user intentions also makes them easier to misuse. It may be easier to use these models to generate convincing misinformation, or hateful or abusive content. Alignment techniques are not a panacea for resolving safety issues associated with large language models; rather, they should be used as one tool in a broader safety ecosystem. Aside from intentional misuse, there are many domains where large language models should be deployed only with great care, or not at all. Examples include high-stakes domains such as medical diagnoses, classifying people based on protected characteristics, determining eligibility for credit, employment, or housing, generating political advertisements, and law enforcement.

Finally, the question of who these models are aligned to is extremely important, and will significantly affect whether the net impact of these models is positive or negative; we discuss this in Appendix G.2.

## Acknowledgements

First, we would like to thank Lilian Weng, Jason Kwon, Boris Power, Che Chang, Josh Achiam, Steven Adler, Gretchen Krueger, Miles Brundage, Tyna Eloundou, Gillian Hadfield, Irene Soliaman, Christy Dennison, Daniel Ziegler, William Saunders, Beth Barnes, Cathy Yeh, Nick Cammaratta, Jonathan Ward, Matt Knight, Pranav Shyam, Alec Radford, and others at OpenAI for discussions throughout the course of the project that helped shape our research direction. We thank Brian Green, Irina Raicu, Subbu Vincent, Varoon Mathur, Kate Crawford, Su Lin Blodgett, Bertie Vidgen, and Paul Röttger for discussions and feedback on our approach. Finally, we thank Sam Bowman, Matthew Rahtz, Ben Mann, Liam Fedus, Helen Ngo, Josh Achiam, Leo Gao, Jared Kaplan, Cathy Yeh, Miles Brundage, Gillian Hadfield, Cooper Raterink, Gretchen Krueger, Tyna Eloundou, Rafal Jakubanis, and Steven Adler for providing feedback on this paper. We'd also like to thank Owain Evans and Stephanie Lin for pointing out the fact that the automatic TruthfulQA metrics were overstating the gains of our PPO models.

Thanks to those who contributed in various ways to the infrastructure used to train and deploy our models, including: Daniel Ziegler, William Saunders, Brooke Chan, Dave Cummings, Chris Hesse, Shantanu Jain, Michael Petrov, Greg Brockman, Felipe Such, Alethea Power, and the entire OpenAI supercomputing team. We'd also like to thank Suchir Balaji for help with recalibration, to Alper Ercetin and Justin Wang for designing the main diagram in this paper, and to the OpenAI Comms team for helping with the release, including: Steve Dowling, Hannah Wong, Natalie Summers, and Elie Georges.

Finally, we want to thank our labelers, without whom this work would not have been possible: Meave Fryer, Sara Tirmizi, James Carroll, Jian Ouyang, Michelle Brothers, Conor Agnew, Joe Kwon, John Morton, Emma Duncan, Delia Randolph, Kaylee Weeks, Alexej Savreux, Siam Ahsan, Rashed Sorwar, Atresha Singh, Muhaiminul Rukshat, Caroline Oliveira, Juan Pablo Castaño Rendón, Atqiya Abida Anjum, Tinashe Mapolisa, Celeste Fejzo, Caio Oleskovicz, Salahuddin Ahmed, Elena Green, Ben Harmelin, Vladan Djordjevic, Victoria Ebbets, Melissa Mejia, Emill Jayson Caypuno, Rachelle Froyalde, Russell M. Bernandez, Jennifer Brillo, Jacob Bryan, Carla Rodriguez, Evgeniya Rabinovich, Morris Stuttard, Rachelle Froyalde, Roxanne Addison, Sarah Nogly, Chait Singh.

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
