# OpenReview forum: "Training language models to follow instructions with human feedback"
_NeurIPS.cc/2022/Conference — NeurIPS 2022 Accept_

### Official Review · Reviewer_mecL · 2022-07-04

**Rating:** 5
**Confidence:** 3
**Soundness:** 3 good
**Presentation:** 3 good
**Contribution:** 3 good

**Summary:**

This work proposes a instruction tuning process for gpt3. It involves three stages: (1) Collect demonstration data, and train a supervised policy. (2) Collect comparison data, and train a reward model. (3) Optimize a policy against the reward model using PPO. It is compared to a few ablation baselines, or models trained on other instruction datasets.
It is compared with a number of strong baselines, and the performance is strong. It is shown that with the instruction tuning, the model can beat a 100x larger LM. (but keep in mind that larger LM is not deliberatedly tuned on instructions.)

**Questions:**

Would one ablation baseline be directly do RL with human feedback ? (without training the reward model)

If you have a smaller size of human feedback, how would the performance be affected?

**Limitations:**

Yes.

**Strengths And Weaknesses:**

Strength:
The collected data would be a good resource for further research into instructed LMs.

The comparison is solid and the performance is strong. It is shown that with the instruction tuning, the model can beat a 100x larger LM. (but keep in mind that larger LM is not deliberatedly tuned on instructions.)

Weakness:
Collecting data/ranking, is of course, costly.

It seems to me that this work is kinda typically adding more data/human labor, and the performance goes up. While the performance is good, the novelty is limited.

More analysis experiment could be done. (see my questions)

---

> ### Comment · Reviewer_mecL · 2022-08-10
> **thanks for the response**
>
> Thanks for the response. The authors did not answer all of my questions. But given that all reviewers are positive about this paper, I'll maintain my score.

---

### Official Review · Reviewer_7bBz · 2022-07-07

**Rating:** 8
**Confidence:** 5
**Soundness:** 4 excellent
**Presentation:** 4 excellent
**Contribution:** 4 excellent

**Summary:**

This paper studies fine-tuning language models to produce outputs thare are more aligned with what users would like them to do. Given the increases focus on language models and their failure modes (hallucinations, toxic responses etc.) this is an important research direction.

**Questions:**

* Is the comparison between the SFT and PPO models fair? SFT was trained on 13K labeled prompts while PPO collectively uses 13K + 33K (both labeled) + 31K unlabeled prompts. Disclaimer: I did not read the appendix.
* Do you truly need a reward model? You could directly train the final model on the 33K PPO dataset using expected risk which uses the labeler rankings as a reward. You would not use the 31K unlabeled prompts, but the result would give you an indication of the value of these unlabeled prompts.

**Limitations:**

The authors adequately addressed the limitations and potential negative societal impact of their work.

**Strengths And Weaknesses:**

+ large scale evaluation in terms of model sizes (GPT-3 size) and number of considered tasks
+ effective approach to improve outputs of LMs with respect to what users expect
+ the problem is clearly significant as language models are an increasingly important research direction

---

### Official Review · Reviewer_bNxf · 2022-07-11

**Rating:** 6
**Confidence:** 4
**Soundness:** 3 good
**Presentation:** 3 good
**Contribution:** 3 good

**Summary:**

This paper addresses the challenge of aligning large language models with user intents. They introduce a three-step process to do so: 1) Train supervised models on (human-collected) prompts and the (human-labeled) outputs 2) Use the models to generate labels which are then ranked by humans according to the desired metrics (bias, toxicity, etc.). Then train a Reward model on the data 3) Fine-tune a supervised policy using PPO algorithm with rewards produced by the Reward model.

They show that the resulting models are better at following user instructions. They make fewer mistakes, use less offensive language, and have fewer hallucinations, as measured by automatic metrics and human evaluation.

**Questions:**

- Is this approach specific to large language models or can it be used for smaller LLMs (~1-10B parameters)?
- What are the possible solutions to avoid manipulation or misuse of alignment by certain individuals? Is it possible to introduce "inhibition" so that models refrain from following user's instructions if deemed biased, toxic, or inappropriate?

**Strengths And Weaknesses:**

Strengths:
- This work takes an important step in reducing bias, toxicity, and other ethical issues concerning large language models.
- The results align with the premises and claims. The discussion provides a good comparison between GPT3 and InstructGPT3 capabilities and where and why InstructGPT3 does not improve over GPT3.
- The paper is clear and easy to follow. It does not overclaim and cites relevant work where appropriate.

Weaknesses:
- This approach relies heavily on human-collected and labeled data. Accessibility of such data, esp. for small research organizations are not clear. Also, the compute power required to fine-tune these models is quite high. As a result, the benefits of this research might not be applicable to the larger research community. The authors do not mention whether they intend to release their dataset or software.
- The approach does not provide a comprehensive solution to the general problem of bias and toxicity. The current approach is mainly data-driven and relies on reward signals which amplifies user preferences, etc. Thus an ill-intended user is now better equipped to mis use such models for their own benefit.
- It's not clear whether current approach can be applied to smaller LLMs (~1-10B parameters).

---

> ### Comment · Reviewer_bNxf · 2022-08-10
> **New comment**
>
> I confirm that I've read the author's response and some of my concerns remain. I'll keep my current score.

---

### Official Review · Reviewer_nCXU · 2022-07-11

**Rating:** 7
**Confidence:** 4
**Soundness:** 3 good
**Presentation:** 4 excellent
**Contribution:** 3 good

**Summary:**

This paper presents an interesting approach to align the language models (GPT-3 in this case) with their users' intent. This is very important because often these language models generate outputs that contain untruthful and toxic information that is not helpful to the user. The paper proposes a three step framework to improve the model’s alignment with its users: (1) supervised finetune of the GPT-3 models with a dataset of labeler demonstrations. (2) train a reward/ranking model using the fine-tuned model and a dataset of labeler annotations comparing model outputs. (3) further fine-tune the GPT-3 model with reinforcement learning using the trained reward model.  This paper has thorough human evaluations on the resulting fine-tuned model (called as InstructGPT). The paper shows that InstructGPT improves on various aspects including truthfulness and reduces toxic output generation in comparison to the baseline GPT-3.

**Questions:**

1) Can you provide some empirical evidence on by how much margin 1.3B InstructGPT is better than 175B GPT-3?
2) Can you provide evidence that there is no contamination of training data and evaluation datasets (NLP datasets) used in the results section?
3) How is the hallucination rate of a few-shot setup in comparison to zero-shot setup (for both GPT and InstructGPT)?
4) It seems that for RM model training, the paper used more than double the data used for SFT. Is there any reason for this? How much minimum data is decent enough for the RM model training?
5) Do you plan to release the data used in SFT and RM?


**Limitations:**

Adequately addressed the limitations.

**Strengths And Weaknesses:**

Strengths:
1)  The proposed approach is very interesting and novel. A similar idea has been applied previously for improving summarization but the current paper addresses a much broader distribution of tasks.
2) The human data collection process is well documented (in the Appendix).
3) The experimental results are very thorough with a lot of human evaluations. Human evaluations make the most sense to show the effectiveness of proposed model to cover a broad distribution of tasks

Weaknesses:
1) The paper provided only minimal information on the deduplication process of prompts, that too in the supplementary. I think it is important to clearly mention the full details of this process. It is not clear how the paraphrased prompts (or prompts with the same example or task but with different templates) are handled.
2) Also, it is unclear whether there is any overlap on the training data and NLP datasets used in the paper for some ablations (e.g., RealToxicityPrompts and TruthfulQA). It would be useful to share some analysis on the data distributions.
3) Sorry if I missed it somehow, but I did not find any “empirical” evidence for the claim: “1.3B parameter InstructGPT model are preferred to outputs from the 175B GPT-3”

---

### Author Response · Authors · 2022-08-02
**Response to reviews**

We thank their reviewers for their kind feedback. We are glad to hear the reviewers find the paper “interesting and novel” (R1), “clear and easy to follow” (R2), and with “very thorough” experimental results (R1) on a problem that is “clearly significant” (R3).

We address concerns of the reviewers below.


1) Performance on smaller (1B-10B param) models (R1 + R2)

R2 states that it is not clear if our method can be applied to 1B-10B param models. We do in fact train 1.3B and 6B versions of InstructGPT, which form a core part of our results. In Figure 1, we show that the 1.3B InstructGPT (labeled ‘PPO’) outperforms the 175B GPT-3 (labeled ‘GPT (prompted)’ ) on human preferences (y axis). Figure 1 also addresses R1’s question about evidence for 1.3B InstructGPT’s performance. In our camera-ready copy, we will edit the caption of Figure 1 to make this result clearer.


2) Cost of human data collection and compute is high (R2 + R4)

We agree that the cost of collecting data and training large language models can be high. But we also believe that our method might make smaller LMs more competitive – given that our 1.3B InstructGPT outperforms a 175B GPT-3 on our API distribution, collecting comparison data and using RLHF might lead to similar performance gains for smaller LMs on other NLP tasks. This would be a significant cost savings, as collecting human data is far cheaper than training a 100x larger LM, and has many other advantages (it’s easier to specify model behavior, improvements in toxicity and truthfulness, etc.).


3) Is the comparison between PPO and SFT fair? (R1 + R3)

It is true that our comparison between PPO and SFT is not fair, as we use way more comparison data than demonstration data. However, the point of our paper isn’t to show that PPO with an RM on comparison data is necessarily the best algorithm to use, but rather to show an existence proof that this method can lead to extremely strong results at a large scale on realistic language tasks. SFT and PPO have been directly compared in a recent paper on book summarization [1], with the finding that PPO was more data efficient.


4) Prompt duplication + overlap with NLP datasets (R1)

We assume R1 is asking about potential overfitting to our eval sets, which is a reasonable concern. We generally did not check for overlap between our training set and our public NLP benchmarks (though we are using most of these to simply measure regressions, and we don’t claim any SOTA results) – the exception for this is TruthfulQA, where we did a very light check (confirmed that a common TruthfulQA question did not appear in our test sets).

Upon further investigation to answer this question, we found that our description of our deduplication procedure was slightly inaccurate (it applied to validation sets that did not make it into the paper) – we don’t check for shared common prefixes, but rather we remove prompts that entirely contains another prompt. We will correct this in our camera-ready version. (Separately, we’ve found that more aggressive deduplication does not significantly affect our results). Our biggest defense against overfitting on our API dataset is that we split our training / validation / test sets by user, so our test set consists entirely of prompts from users that were not trained on. Empirically, we believe significant overfitting is unlikely as InstructGPT can generalize to tasks that are extremely rare in the training dataset (see Sec 4.3).


5) Avoiding misuse (R2)

We agree that InstructGPT can make it easier for malicious actors to misuse LMs, and we note this in Section 5.3. We believe mitigating this requires interventions at many levels of the model deployment pipeline; for example, a process for approving applications before they go into production, and tools to monitor and react to misuse. However, some of these are not enforceable if the model is open-sourced. At the model level, one could use our method to fine-tune the LM to refuse to respond to certain tasks that may be harmful (using a combination of demos + comparisons). This is an exciting direction for future work.


6) Do you plan to release the SFT and RM data? (R1 + R2)

Unfortunately, since the majority of our data comes from customers using our API, we are not able to release it publicly.


[1] Wu et al., “Recursively Summarizing Books with Human Feedback”, 2021.

---

### Meta-Review · Area_Chair_44R8 · 2022-08-25

**Recommendation:** Accept
**Confidence:** Less certain

**Metareview:**

This paper demonstrates that a smaller model (1.7B InstructGPT) trained with human feedback (using a learned reward model and PPO) is competitive with a much larger pretrained model (175B GPT-3). The results are clearly convincing. The main downside is that there isn’t much analysis on the proposed method (e.g. the influence of the RL algorithm and the amount of human feedback needed). That said, the approach is novel and the results are impressive, which points out new directions for collecting better pretraining data. Thus, I lean towards acceptance.

**Award:**

No

---

### Decision · Program_Chairs · 2022-09-14

Accept